# Ontological Model in the Identification of Emotional Aspects in Alzheimer Patients

**DOI:** 10.3390/healthcare11101392

**Published:** 2023-05-11

**Authors:** David Ricardo Castillo Salazar, Laura Lanzarini, Héctor Gómez, Saravana Prakash Thirumuruganandham, Dario Xavier Castillo Salazar

**Affiliations:** 1Centro de Investigación de Ciencias Humanas y de la Educación (CICHE), Universidad Indoamérica, Ambato 180103, Ecuador; 2Facultad de Informática, Universidad Nacional de la Plata, La Plata 1900, Argentina; 3Instituto de Investigación en Informática LIDI (Centro CICPBA), Facultad de Informática, Universidad Nacional de la Plata, La Plata 1900, Argentina; 4Centro de Posgrados, Universidad Técnica de Ambato, Ambato 180104, Ecuador; 5Unidad Educativa Las Americas, Area de Ciencias Experimentales, Ambato 180103, Ecuador

**Keywords:** Alzheimer, ontology, Protégé, Pellet Reasoner, semantics, taxonomy, apache netbeans

## Abstract

The present work describes the development of a conceptual representation model of the domain of the theory of formal grammars and abstract machines through ontological modeling. The main goal is to develop an ontology capable of deriving new knowledge about the mood of an Alzheimer’s patient in the categories of wandering, nervous, depressed, disoriented or bored. The patients are from elderly care centers in Ambato Canton-Ecuador. The population consists of 147 individuals of both sexes, diagnosed with Alzheimer’s disease, with ages ranging from 75 to 89 years. The methods used are the taxonomic levels, the semantic categories and the ontological primitives. All these aspects allow the computational generation of an ontological structure, in addition to the use of the proprietary tool Pellet Reasoner as well as Apache NetBeans from Java for process completion. As a result, an ontological model is generated using its instances and Pellet Reasoner to identify the expected effect. It is noted that the ontologies come from the artificial intelligence domain. In this case, they are represented by aspects of real-world context that relate to common vocabularies for humans and applications working in a domain or area of interest.

## 1. Introduction

This manuscript is a research study. The result is an ontology based on artificial intelligence insights based on posture estimates. The emergence and spread of diseases in the last decade has been a source of concern for humanity, especially in the field of health sciences. Alzheimer’s disease is one of the diseases that disproportionately affects the elderly worldwide. Available pharmacological and non-pharmacological treatments have slowed the deterioration and decmidrule of patients’ health and affected their mood and emotional state. The rapid and improved development of technology for early diagnosis of Alzheimer’s disease has led to the use of artificial intelligence through the use of specific tools such as Protege to generate reasoned results that can be tailored to the patient’s mood. The inference of the Pellet Reasoner has enabled the creation of ontologies for the relationship between classes and subclasses of descriptive logic to provide specific results related to a positional state to determine the mental states of people. An aspect that allows psychiatrists to determine the degree and level of care for Alzheimer’s patients based on their emotional aspect. In the article developed by [1], the importance of information systems in the communication processes of people is generalized, as an example can be manifested speech, facial expressions, which are considered the basis of human emotions. The proposed ontology focuses on the transmission of emotions, in particular on the communication process between the people who use the system and the interactive part; the study focuses on relevance indicators used to mediate the different affective states, where it was possible to include ontologies and of which platforms were designed and created to guide their development with emotional resources available to those who use the interfaces. higher nature. The result of the recognition performed is a personalized process in which individuals teach the system their own behavior and the way they express their emotions. Therefore, while adapting the system to living beings is feasible, using a single session to collect the necessary data for the interpretation process with a high degree of reference accuracy to the conversations conducted with patients may not be practical.

The author [2] proposes a model for measuring sentiment based on a domain ontology that builds a fine-grained formality based on the characteristics of the complex product, which takes advantage of the method based on the quantification of emotional intensity, with objectivity developed to create a taxonomy of the same type from the existing precursors on the web, which can have high precision and interpretability. The analysed model can be useful in individual or collective decision making and provide corrective information for online review portals.

In the research developed by [3] an ontology of records related to affectivity is proposed. It allows modeling information about the origin and meaning of data represented in time series, which in this case represent the emotional states of features derived from biosignals. In summary, it is a stand-alone tool that, in its current version, can be used to formally describe a wide variety of data sets.

The study of [4] is concerned with the contexts related to the perceptions and decisions that people express in certain situations and that have their origin in emotions. To this end, an ontology is a way of representing the concepts of domain and relations. An ontological model called Visualized Emotion Ontology (VEO), created in Protégé, is used to express the semantic definitions and visualizations of 25 emotions based on published research. The ontology contains a total of 126 classes, 11 object and data properties, and 25 instances. VEO showed better machine readability (z = 1.12), linguistic quality (z = 0.61), and domain coverage (z = 0.39) compared to a sample of cognitive ontologies. Recently, a computational study of ontological data was published [5]. The use of semantics based on fuzzy ontologies is demonstrated by first considering a separate standard ontology in each input source and then merging the developed ontologies into a unified ontology. In this context, by using a methodology standard and fuzzy logic to support the fuzzy ontology-based system to help clinicians accurately retrieve all required patient data from scattered locations with natural language queries, it is possible to extend the integrated ontology into a fuzzy ontology.

In the research developed by [6,7], it is pointed out that ontologies are the fundamental part of the semantic web due to their representation of knowledge by computer equipment. The research enables the use of VisCOnto, a visualization tool for complex ontologies that creates sub-ontologies in conjunction with WEBVOWL to create meaningful information visualizations by leveraging the semantic inference capabilities that ontologies provide about human behavior and character.

In their work on biomedical ontologies, the authors [8,9] the use of semantic knowledge representation in a specific domain of Alzheimer’s disease. This context and their knowledge exchange enable the modeling of criteria regarding the disease from an ontological perspective. The ontology of Alzheimer’s disease follows a life cycle based on Protégé as a tool for the use of the ontology language. The results show that the ontology integrates and supports scientific search in texts related to 72% when it comes to the recognition of entities from the biological domain.

The research conducted by [10] presents a context related to the development of an ontology that encodes the specific knowledge of a domain of pharmacological treatment related to dementia agitation, where a large amount of data and the use of artificial intelligence technologies allow the use of the ontology to encode the clinical. That is, managing the pharmacological part in a machine-readable format. The Neon method is used with an adapted method consisting of 569 concepts and 48 object properties aligned with biomedical ontology standards. This provides a knowledge base for the development of intelligent systems for the treatment of dementia-related agitation.

The work developed by [11] focuses on the use of Alzheimer’s methodology and application with the use of omic technologies to understand diseases using disease maps to avoid ambiguities and misinterpretations in data. Therefore, the use of ontologies through their axiomatic definitions and logical inference properties is the most appropriate framework to overcome this limitation. Induction to the Discrepancy Map Ontology (DMO) based on concepts from systems biology applied to Alzheimer’s disease and related to redundancy, naming, coherence, process classification, and relationships between pathways demonstrate a strengthened model in The Ontology of Alzheimer’s Disease.

The author [12] presents an approach based on deep convolutional neural networks for the Alzheimer’s disease diagnosis procedure to detect the early stage of the disease (AD). In the biomedical context, little research has been done on aspects related to knowledge related to ontology, they are complex processes that take a lot of time, the research applies approaches based on machine learning with an accuracy of 92.12% and those based on deep learning of convolutional neural networks. A precision level of 94.61% was achieved. The main result of this ontology. The results show that building ontologies using deep learning insights can achieve better results in terms of robustness and scalability.

The article developed by [13] takes a generalization toward biotechnology and computer science, focusing on biomedical data in databases. It is important to focus on building a formal conceptual model that uses semantic design tools with data from specific pathologies such as Alzheimer’s disease with specific data from the Alzheimer’s Disease Neuroimaging Initiative (ADNI), in this case using Big Data and Deep Learning with new diagnostic biomarkers in this type of disease. The results show the development of an ontology that simplifies the knowledge of Alzheimer’s disease and provides new insights.

In research conducted by [14], insight into the AlzPathway disease map is provided, revealing details of the pathophysiology of Alzheimer’s disease. However, in certain cases, certain errors may occur. Therefore, the use of ontologies helps to overcome these kinds of limitations in the context of axiomatic definitions and logical reasoning to transform the Alzheimer’s disease map into an ontology that opens the possibility of using elements from other resources as a generic structure. This text [15] refers to two viewpoints, the first focusing on medical aspects related to Alzheimer’s disease and the second exploring clinical decision support systems based on ontologies, robotics, and mobile applications in the neurological context of Alzheimer’s disease. Therefore, in a sense, the study provides solutions for research in the fields of medicine and engineering, leading to promising results in the precision index in the use of ontologies to process, analyze and establish a semantic relationship of Big Data.

The study conducted by [16] focuses on the use of an ontology as a basis for the analysis of mild cognitive impairment (MCI) in patients with Alzheimer’s disease (AD). Data are available in different formats and technologies, which makes them difficult to process. Therefore, ontologies are presented as a solution for data representation and reuse. In this case, four domains are integrated, related to neurodegenerative diseases, diagnostic tests, cognitive functions, and brain domains. The neurocognitive ontology enables the integration of this knowledge into interoperability and facilitates access to data from different disciplines. This ontology has been validated in scenarios with different use cases.

Another related article is that of [17], which shows the relationship between the structures of knowledge about Alzheimer’s disease (AD) and semantic reasoning through the use of technologies in the biomedical field. The Alzheimer’s Disease Ontology (ADO) described in OWL with clinical, preclinical, experimental, and molecular mechanisms and the Alzheimer’s Disease Map Ontology (AD Map Ontology-ADMO). This provides a response to the complexity of the pathophysiological processes of AD.

This article [18] presents an ontology developed for direct home care of Alzheimer’s patients (AD), assuming that direct caregivers are family members or nurses. An ontological model containing first-level categories, concepts, and relationships for informal caregiving of people with AD has been evaluated in real cases. The study by [19] shows that the approach proposed here is not only viable in emotion recognition, but also shows promise in diagnosing depression in AD and mild cognitive impairment. The previously mentioned work shows the importance and usefulness of ontological contexts for detecting emotional aspects in AD patients. In all this work, methods and software are used to find solutions for different behaviours where scenarios are formed in different human functions.

The present study focuses on the design and structuring of an ontology in which the semantic context of Alzheimer’s patients is identified. This criterion can be understood as the environment in which the elderly develop. One of the most important features is that elderly people can adopt different behaviors depending on their location. To analyze the problem, it is necessary to mention as a cause that mood and anxiety disorders are multifaceted. The combination of genetics, changes in hormone levels, environmental factors, important or unexpected changes in life, and stress seem to be responsible. The study of this social problem in a medical context with the identification of the wandering, nervous, depressed, disoriented, and bored state of mind in elderly patients of both sexes with neurological diseases, in this case Alzheimer’s disease, leads to an interest in the incorporation of computer technologies in the field of artificial intelligence with a focus on machine learning, as well as the state of the study is relevant to the field of academic science. The study is more relevant to the academic field than to the scientific field because it reflects the current state of the science.

This manuscript study is organized as follows: Section 1 describes the relevant research within the developed line of research. Section 2 explains the methods and materials used in this work. The process of information collection and the proposed model related to the development of an ontology for patients with Alzheimer’s disease, in this section of the study, the results of the ontological data are presented in detail.

Scientific studies have made it possible to detect Alzheimer’s symptoms early, processes in which artificial intelligence has been used. In this sense, ref. [20] autonomous learning techniques were used to develop a learning algorithm based on the collection of blood samples from which early Alzheimer’s features were determined. This process took several years to obtain the quantity and quality of information needed to develop a potential prediction algorithm. The study under investigation aims to facilitate diagnosis for internal medicine physicians.

In the same context, research conducted by [21,22] uses artificial intelligence to detect early Alzheimer’s disease in the neurological domain, for which Deep Learning was used by implementing a neural network-based classifier. The convolution, which uses a screening of different cases of magnetic resonance imaging with a size of 128 × 128 mm, allows the discrimination of a dementia state compared to healthy people.

Artificial intelligence is used for the detection of observational backgrounds that have nothing to do with Alzheimer’s disease. This is the detection of Parkinson’s disease, as shown by [23], where a learning algorithm was used to assess motor skills using a battery of standardized exercises. In the same vein, [24] Used artificial intelligence algorithms to develop a logistic regression model for early detection of people with diabetes. A decision tree and vector support machines were also used for this purpose, and it was possible to define ranges in which estimates of 0.613 to 0.727 of the area under the curve were most informative in determining a person with diabetes.

Previous studies have shown that the presence of Alzheimer’s disease can be detected early by applying artificial intelligence from the fields of neurology and internal medicine, whereas this study computationally modified the basic Damerau-Levenshtein distance algorithm by using a programming class referred to in the code as CMP that contains a standard Qt sequence (CC: wandering; CS: nervous; SC: nervous; SS: depressed; CP: disoriented; PC: disoriented; PS: bored; SP: bored; PP: nothing ) to detect mood in Alzheimer’s patients.The study represents an innovative process for the field of health sciences and, in particular, for the field of psychiatry, as the professionals involved in it will be able to make a safe and rigorous diagnosis of the mood of Alzheimer’s patients with quality, speed and reliable data analysis, and to find an effective medical solution for this disease.

In addition, for the predictive factors, the study first assumes the type of mood the AD patient is in, as the likely response variables may have to do with confusion, anxiety, boredom, wandering, and depression. The patient’s ability to stand, walk, or sit also makes the pattern an input variable for the study.

## 2. Materials and Methods

This section clearly describes the techniques used in the study and the materials used to develop the ontology. The ontological structure allows to draw conclusions about the emotional behavior of the patient. The method used focuses on the development of the semantic structure and begins with the study of a taxonomic, semantic and ontological process. The initial topic studied refers to the taxonomic processes, which are considered as a classification system that allows grouping a set of elements within certain predefined categories, as is the case with the species of living beings. References [25,26] states that taxonomies are important in real-world applications in various contexts that serve to represent knowledge that must necessarily be organized with taxonomies in the face of new concepts. The initial topic studied refers to taxonomic processes, which are viewed as a classification system that allows a set of elements to be grouped within certain predefined categories, as is the case with species of living things. In [27], it is stated that taxonomies are important in real-world applications in various contexts that serve to represent knowledge that must necessarily be organized with taxonomies in the face of new concepts. Another important parameter is the ontology, which is based on a set of representative primitives that can be used to model a knowledge domain.

The study’s analysis method was descriptive in nature, allowing an analysis of the specific characteristics of the Alzheimer’s patients, which allowed a classification process in those people with different detected mood states; in this context, the set of digital tools such as protege, apache netbeans, and PCA allowed the creation of an ontology, which welcomed the insertion of statistical inferences for data analytic purposes based on its results. in psychiatric clinical diagnosis of mood states in Alzheimer’s patients. Furthermore, the study employs a predictive method to ensure that axioms are followed because theorems were utilised to describe the relationships between the parts that comprised the ontology. In [28,29] It is obvious that to create an intelligent system, an ontology must be chosen that is based on symbols that represent unique and significant concepts within the domain during the execution process. This symbolism must be used after the domain knowledge has been represented.

Initially, the research process focuses on a set of data stored in an electronic sheet that contains several columns (attributes) such as start, processing time, pose, state, scene. This information comes from videos with an estimated duration of 1 min of 45 Alzheimer’s patients, being men aged 75 to 86 years and women aged 75 to 89 years. The developed method required special equipment consisting of a portable computer with sufficient capacity in CPU for data processing under the Windows operating system. Protégé 5.5 was used to develop the ontology, using the main classes: Patient, Patron, Mental State were connected to the Pellet Reasoner. To migrate the information from Excel to the ontology, Netbeans 12.3 was used, where programming functions were applied to upload the spreadsheet document and load the current ontology to generate the new ontological context, ending with the download of the file OWL. The following section presents the development of the model for the present research. The model shows the development of the ontology capable of deriving new knowledge about the mood of an Alzheimer’s patient in one of the following categories: wandering, nervous, depressed, disoriented, or bored, depending on the context in which the patient finds himself or herself The architecture of the ontological process is shown in Figure 1.

### 2.1. Taxonomy

The study begins by examining the taxonomic process using the taxonomic levels in [30,31]. This is an ordering system that allows a set of elements to be grouped within certain predefined categories. Ref. [32] states that a taxonomy exists when semantic concepts are given an order. This ordering then means a procedure that facilitates classification in a formal analysis using classes and subclasses. Figure 2 graphically represents the taxonomic context, the first level, which refers to Alzheimer’s patients in their real environment.

### 2.2. Alzheimer’S Hierarchical Taxonomy

The design and elaboration of the hierarchical taxonomy of Alzheimer’s disease are the result of the fieldwork carried out by the researchers, which consisted in personally visiting fifteen gerontological centers, where the behavior of one hundred and forty-seven selected patients was observed, who had a medical certificate stating that they suffered from Alzheimer’s disease. During the fieldwork and observation, eleven experts in the field of psychiatry collaborated and, based on their extensive experience, established their criteria for determining seven classes of descriptive logic of observation, of which the thirty-six subclasses are of the same type (descriptive logic). The classes and subclasses of descriptive logic used in the study based on the experience of psychiatrists are described in Table 1.

### 2.3. Semantic

In its simplest form, semantics is a word that has a common purpose. In this study, certain major categories are defined, and their classes and subclasses are used in the context of the patient’s environment in [1,33]. Relationship of an ontology in an existing environment based on various parameters, this context model is based on an ontology that defines the general concepts of the mentioned study. The components of the semantics are shown in Table 1.

An ontology is the set of representative primitives that can be used to model a domain of knowledge in different domains of cognition, as described in [15,34,35]. Also in [36] it is pointed out that the primitives represented symbolically are typically classes (sets), attributes (properties), and relations (relationships between class members). In a deeper context, it is possible to identify specific attributes. In the developed research the relation between classes pattern and mood is defined, which are part of the concept of semantic patterns, as shown in Table 2.

### 2.4. Protegé

Ontology definition tools are software applications [21] that allow the encoding of an ontology based on the Protégé language. Protégé is a free and open-access working platform that provides myriad tools for elaborating domains as well as applications for basic knowledge about ontologies [37].

### 2.5. Ontological Structure

The ontological structure defined by classes, object properties, quantification and cardinality constraints, and instances is shown in Figure 3, where the working environment of the Protégé tool is also described. In [23,33,38] domain ontologies show that the definitions of properties and classes are fixed. Binary relations between elements are found in living things as well as in fundamental concepts. The objects in different ontological domains can be located within classes, the attributes, classes, and induction of relations are given by axioms, the derivation of this principle within the ontology is called a terminology, and the applications of a particular form are called domains.

Figure 3 shows the six classes and twenty-four subclasses that emerged from an analysis of the postures of Alzheimer’s patients in gerontological centres, and the instances that relate to each other to create the detailed Ontologa Graphica (OntoGraf) shown in Figure 3.

### 2.6. Apache Netbeans-Java

Once the classes, their properties and relationships are defined, we work with the software tool Apache NetBeans 12.3, a Java programming language, in such a way that the programs created in it can be executed on different types of architectures and computing devices without any modifications. In [39,40], we talk about the importance of using multiple languages to leverage the strengths of each language. For example, applications can be developed in Java to solve various problems in a computational context. As shown in Figure 4, the execution of the application that allows the creation of a graphical interface developed in Java code, as well as a working environment that is a friendly and easy to use area. In [41,42], the use of semantic web technologies such as ontology, Jena API and SPARQL queries can be reused with the data in all applications, whatever can be discovered or extracted to obtain specific information needed by the users of this application.

### 2.7. Evaluation

In Alzheimer’s disease, for example, the study could assess net activation of postures as a result of cerebral mental state, recognise the stage the patient is in in relation to the disease, which could be early, intermediate, or late and classified as dementia, or assess the stages of Alzheimer’s disease in other ways that must be equally weighted by the subjectivity of psychiatry. In this case, the knowledge representation and algorithm could be adjusted to reflect the patient’s functional mental state and level of PD.

Since the patient’s body movements are required to determine the position and later the mood, the system developed in the study could detect the mood in Alzheimer’s patients when the disease is in a mild stage.

The study has a positive control in the sense that the awareness of what is measured by the system in terms of mood has similarities to the criteria established by experts in the field of psychiatry, and a negative control in the sense that false positives may occur if the quality of the information feeding the learning algorithm is not achieved.

## 3. Results

### 3.1. Pellet Reasoner

The Alzheimer’s Disease Ontology (ADO) encompasses a broad range of fundamental concepts and is intended to provide particular insights into Alzheimer’s disease. The heterogeneity modeled in this semantic framework attempts to link all the major concepts of the study as primary classes of the ontology to the more specific concepts that are part of this structure. The primary elements (Root Ideas) of the ontology include features of knowledge about Alzheimer’s disease such as scene, condition, patient, and pattern. Each of these classes has its own subclasses. The ”State” and ”Pattern” subclasses contain terms that have contributed significantly to the knowledge of Alzheimer’s patients. Currently, much of Alzheimer’s research focuses on preclinical studies, which are usually conducted in elderly patients with the disease. In this study, the Pellet 2.2 reasoner was used because it has some features that other reasoners do not, such as the ability to work with complex data types and processing rules. In [43], we develop an ontological model for caring for people with dementia and a reasoning system that adaptively produces care guidelines under different circumstances. In this sense, ontological models have important properties when it comes to capacity in terms of inference and computation time. In [44], the importance of using the Pellet, Jena, and Protégé tools.

The programme and the described tools generate the interpretation in the classes as an equivalent element for the subclasses, and the classes connected to the superclasses generate a crossing process.

Table 3 shows the conceptualizations chosen for the study to identify the mental state of Alzheimer’s patients from the epistemological subjectivity of the treating psychiatrists in the gerontological centres, who conceptualised the mental state Disoriented, Nervous, Bored, Wandering, Depressed in their own words.

The study presented begins with the recording of data that, from the recording of videos of Alzheimer patients in care centers, subsequently processed digitally in the software developed for the estimation of semantic pose based on the fundamentals of artificial intelligence, generates results, as shown in Table 4.

In Figure 5a it can be seen that the longest arrows of Dim1 are indented and nothing corresponding to the highest coefficients for this component. The Dim2 component, on the other hand, shows a disoriented variable with greater length; 56.8% of the variation generated is explained by the Dim1 and Dim2 components. In Figure 5b Dim2, the highest coefficient is disoriented and in Dim3 migrates the one containing the largest length; 52.7% of the generated variation is explained by components Dim2 and Dim3. In Figure 5c, the extent of the arrows shows that depressed and nothing have the highest coefficients, while in the other component migrating is shown as the component with the largest length, 56.1% of the variation generated is explained in Dim1 and Dim3. The present study shows in Figure 6, the results of the Alzheimer’s patients in a three-dimensional diagram, where the representations can be observed by the arrows in different colors associated with each of the variables. The variables are associated with positive values in PC1 (principal components) and PC2. References [43,45] The length of the arrows indicates how much each variable contributes to the calculation of each principal component: Disoriented, Wandering, Depressed, Bored, Nervous, No observation.

Next, Figure 7 shows that the categories for the variables No Observation = 30.1%, Disoriented = 26.7%, Wandering = 26%, Depressed = 17.1%, Bored = 0%, Nervous = 0%, reach percentages. From the conducted study, it can be seen that in the activities that Alzheimer’s patients usually perform, the highest percentages are found in the categories of disorientation, wandering and depression. This means that the orientation of the disease is related to its progressive course and that the activities they perform daily are more pronounced in these parameters.

An ontology is conceived as a set of primitives that model a particular domain of knowledge. Representative primitives are essentially classes (sets), attributes (properties), and relations (relationships between class members). As for the classes of the ontology named Alzheimer, it is structured by the classes: scene, state, patient, pattern and these in turn with their respective subclasses and a comment that is referenced in each case.

Using Apache Netbeans as a software platform, an application is developed that creates a graphical user interface developed in Java code.

The new ontology generated in Netbeans with the Java code enables the identification of the instances of the subclasses belonging to the Scene and State classes, among others, as shown in Figure 6b the instances of a subclass that is an instance of a superclass. In [16,46], the entire process of ontological construction is described, starting with data collection and identification of the main concepts of the ontology in terms of classes and subclasses. Several existing ontologies describe a subset of the vast domain of human diseases known as symptom ontology. According to [3,36], ontologies consist of objects (or individuals), properties (binary interactions between objects, often called roles), and classes (or concepts); objects can be instances of classes. In domain ontology, the definition of attributes, classes, and the relationships between them by axioms is taken as foundation.

When performing analysis with greater formality, it is worth pointing out that when creating ontologies, one organizes sets of In a general context, the number of moments ranges from a few hundreds in the classes to tens of thousands in the detailed elements, although the number of moments in quantities of greater value might range from hundreds to hundreds of thousands. Using an ontology can be managed at these scales, populating an ontology is not something that can be done manually.

Instances are often obtained from databases, but there are other sources and techniques, such as automatic extraction of text instances. In this case, it is important to know that the information comes from an Excel file created by a machine learning algorithm. The pellet reasoner used allows the processing of complex data types and the processing of rules under the pellet environment took a few seconds to analyze the full information of the ontology. At the beginning of this process, the reasoner checks the consistency of the knowledge base using logical resolution methods that act on the stored axioms and provide satisfactory results. Figure 8 shows the use of pellet.

Reasoners such as Pellet, Fact++, RacerPro, Hermit, and others are used in [27] to derive hidden information from ontologies. Pellet is considered the best reasoner in terms of logic, usability, practicality, and performance. The Jena semantic framework, on the other hand, has a query engine. As a result, the criterion supports research inference tools, assuming that Jena and Pellet allow working with unification and pruning classes. In [18] it is mentioned that the main service of a reasoner is to investigate whether a class is a subclass of another class or not. A reasoner can compute the class hierarchy derived from an ontology by performing these checks on the classes. The reasoner can determine whether a class can have instances or not based on its description (conditions).

The OntoGraf view class allows you to navigate interactively between OWL ontology relationships that support the management of multiple layouts to automatically organize the ontology structure. They allow for the integration of different subclasses with net attributes based on the real object and its equivalent. The established relations, as well as the various nodes, have the ability to be filtered in order to produce a desired view.

The OntoGraf view class allows you to navigate interactively between OWL ontology relationships that support the management of multiple layouts to automatically organize the ontology structure. They allow the integration of different subclasses with network attributes based on the real object and its equivalent. Both the relationships created and the different nodes can be filtered to get the desired view. Figure 9. According to [18], the OntoGraf representation supports interactive navigation through ontology relationships, presenting several designs for the automatic organisation of the ontological structure, thus supporting the different relationships: subclass, individual, domain/range object properties, and equivalence. In order to capture knowledge, an OntoGraph of the relationship between disease and symptoms is given from cardiac diagnostic records, books, and in partnership with medical specialists [16].

In this context, the model in Figure 9 includes the concepts and attributes related to the study that relate to some good ideas of ontological modeling related to the mental health of a person with Alzheimer’s disease, which has various factors classified into classes and subclasses, for your better understanding; these are known as ontologies and use a OWL schema. The details of each part of the schema are.

### 3.2. The OWLThing: From Which the Whole Scheme Descends

#### 3.2.1. The Patient Class

Which is intended for patient information and has its subclasses to satisfy their information needs. which are:

Habit: are the habits of the patient.Livelihoods: are the means of your daily subsistence.Town: the location of your home.Culture: ethnicity or race of the patient.

#### 3.2.2. The Scene Class

Where events happen that have only one of the following characteristics or subclass:

corridor.courtyard.dining room.bathroom.rooms.physical therapy room.electrotherapy room.street, bedroom.crafts room.garden.recreation room.

### 3.3. The Pattern Class

Patients have activity patterns that are inside the pattern class and have one of the available subclasses, which are:

walking: patient walking action.sitting: patient sitting action.standing: action of the patient standing.

### 3.4. The State Class

Patients also have an emotional state, which is within subclasses and can only be 1, which are:

Bored.disorientated.depressed.wandered.highly strung.

The State class is related to the patron class, through the at-least-one object property, patients link one of their emotional states with:

bored with sitting and standing.disoriented with sitting and standing.depressed with sitting.wandered with walking and must be repeated at least 4 times.nervous with sitting and standing.

This scheme described above in the form of an ontology describes the complexity of the factors that make up a patient so that it can be understood more easily.

## 4. Discussion

Ontology is a discipline that cultivates philosophy. While it is a productive area within the contemporary research system, like any method of inquiry it has its limitations. The scope of this research environment is currently focused on the field of health sciences, in the discipline of psychiatry, to determine the mental state of people with Alzheimer’s disease in a mild stage, for which artificial intelligence was used. The study is similar to that of the author [22], who also integrated ontology with artificial intelligence through the applicability of neural networks generated within working systems and interactivity with technology, for the development of the study the distance algorithm compared to the study of [23], Levenshtein, in which new algorithmic elements were introduced that made it possible to determine different postures of people with the aim of detecting the state of mind, the ontologies created represent a design for handling postures with the aim of determining the state or progression of various diseases such as Parkinson’s and Alzheimer’s itself.

The study is similar to the author’s [22] in that the research used three-dimensional movies to detect vector position to determine the mental status of people in the field of psychiatry, while [22] used three-dimensional movies to determine the degree of Alzheimer’s disease in the clinical setting. In this context, the study used a broadcast camera to record the different positions of individuals, while the [22] Person study used 3D magnetic resonance imaging of the brain. The graphical representation of the ontology (OntoGraf) allows visual assimilation of what is learned and an understanding of the structure between classes and subclasses, allowing information processing to be guided, communicated, and modelled to discover the relationships associated with classes and subclasses. The above is in agreement with author [28] who states that for the development of an intelligent system, symbols must be selected to represent different and important concepts within the domain during the execution process, using arrows and other elements such as continuous lines that are part of the general ontology. Once the domain knowledge has been represented, this symbolism must be used.

## 5. Conclusions

In this work, the following properties are observed and evaluated accordingly using Protege software and PCA analysis.

1.The study develops an ontology-based model for detecting the mood of patients with Alzheimer’s disease, promoting standardized, consistent, comprehensive, and highly accurate means for mood representation of patients with the previously exposed disease [47].The development of ontology in the field of health sciences and specifically in psychiatry, provides artificial knowledge-based diagnosis to reduce the time and specify the diagnosis of the mood of patients with Alzheimer’s disease, where the stages of taxonomy and semantics are the pillars for the construction of patterns in the identification of movements and their subsequent analysis to evaluate and diagnose the mental state of the elderly.2.The interoperability of semantics in the development of the mapping of the patient’s mental information takes the artificial knowledge and is represented by means of a computational format by machines using the ontology. The approach of estimating the computational pose, as detailed in this research, can represent and become an essential tool in translational computational processes, which may lead to various applications in the future to support the decision making of psychiatrists treating people with AD.3.The ontological architecture of the model for recognising the mood of Alzheimer’s patients begins with the reality of the world in which older people with this disease live. The taxonomy emerges from the context of the words describing the coexisting habitats of the patients, which leads to the establishment of semantics, the stage at which classes and subclasses are formed.The axiom adaptations constitute the motor organ of ontology for the study of Alzheimer’s patients’ mood, such that the established logical-mathematical links lead to the adaptation of algorithms for the detection of movements. of patients in the development of their everyday activities.4.Accordingly, Alzheimer’s patients represent the original class and subclasses (Disoriented, Anxious, Bored, Wandering, Sad) in this context. According to the study, the class patterns (Walking, Standing, and Sitting) are associated with the subclass State, which supports the ontological modelling.5.At the level of software, computational ontology has identified the relationship between pattern class and mood under an object attribute that at least generalises whether or not the patient is moving.6.The results for diagnosis related to the mood of patients with Alzheimer’s disease are obtained with a high degree of efficiency through the application of technologies using the Protege, Pellet tools, and the PCA analysis, allowing the processes recorded from the environment of social development vision developed by machine learning, which accurately determines the actions of older adults.

## Figures and Tables

**Figure 1 healthcare-11-01392-f001:**
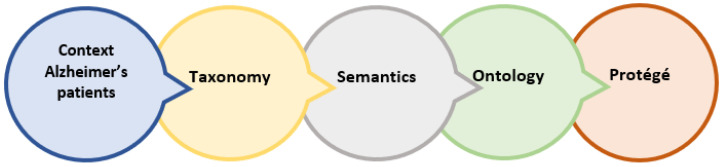
Ontological architecture.

**Figure 2 healthcare-11-01392-f002:**
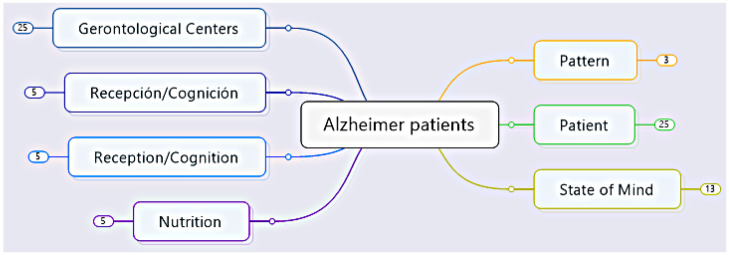
The architecture of the ontological process of Alzheimer’s Taxonomy.

**Figure 3 healthcare-11-01392-f003:**
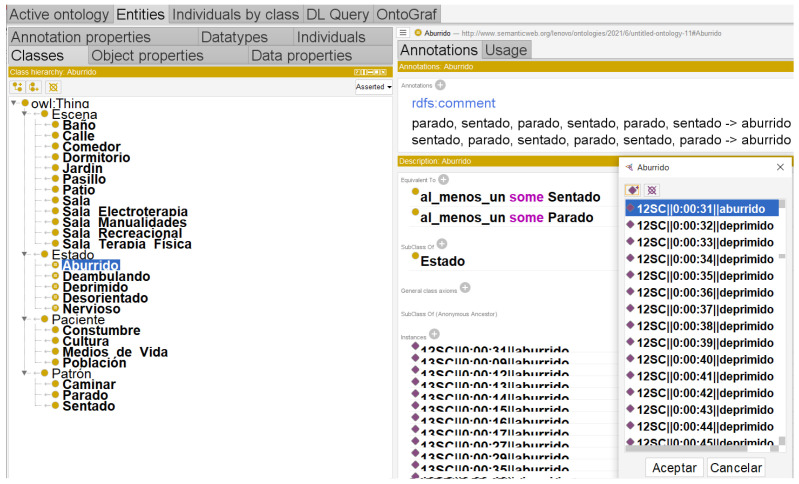
Environment protegé.

**Figure 4 healthcare-11-01392-f004:**
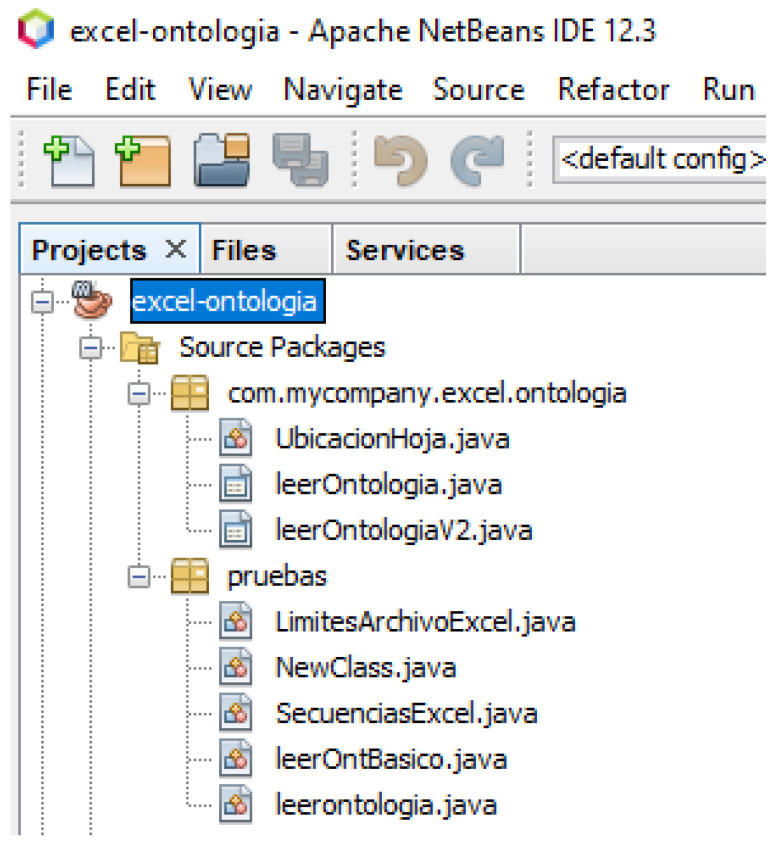
Environment Protege-Netbeans.

**Figure 5 healthcare-11-01392-f005:**
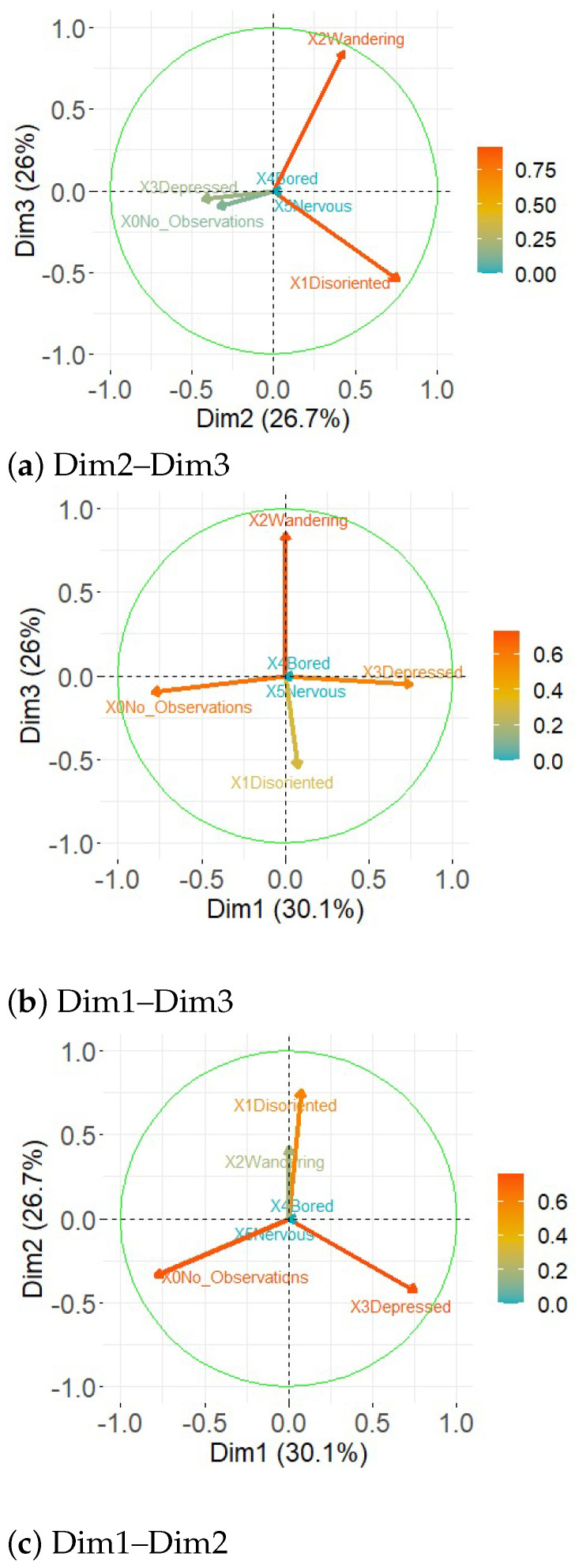
Principal component analysis.

**Figure 6 healthcare-11-01392-f006:**
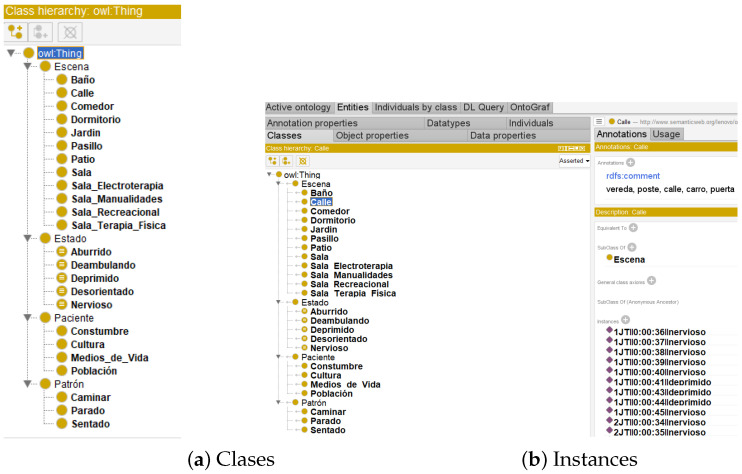
Ontology Class-Subclass.

**Figure 7 healthcare-11-01392-f007:**
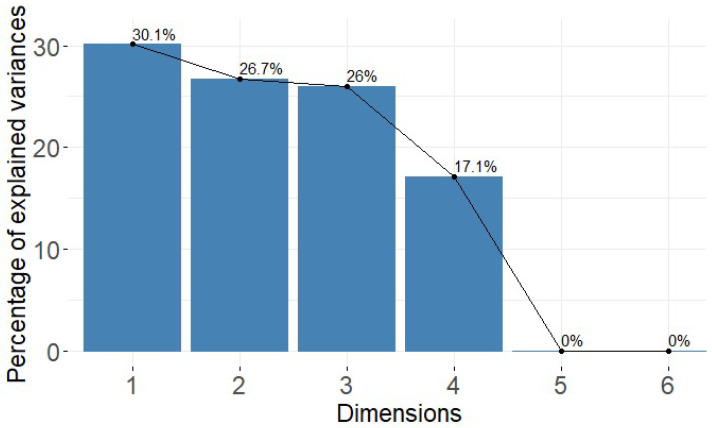
Histogram of Variable Percentage.

**Figure 8 healthcare-11-01392-f008:**
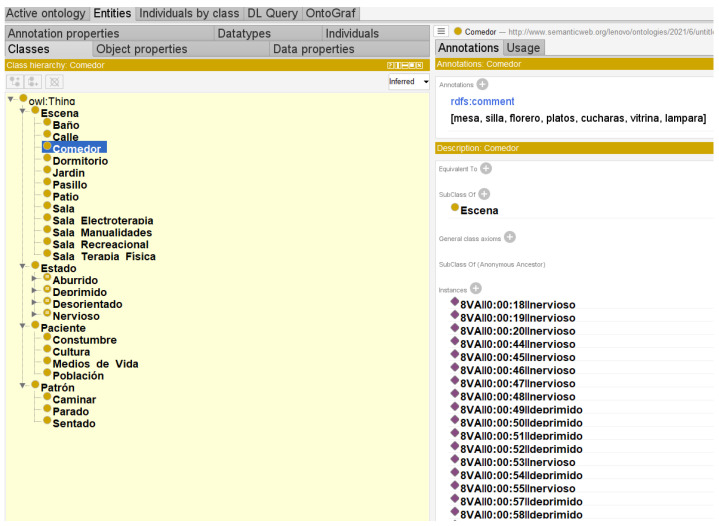
Protégé Pellet.

**Figure 9 healthcare-11-01392-f009:**
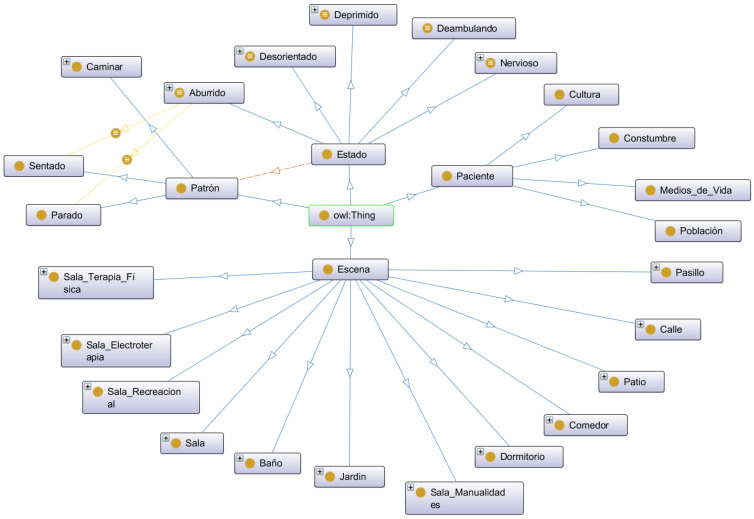
Ontograf of elderly patients on their mood description.

**Table 1 healthcare-11-01392-t001:** Hierarchy of classes and subclasses.

Class	SubClass
Patient	Man
	Woman
	Ethnographic Data
State of mind	Disoriented
	Nervous
	Boring
	Wander
	Depressed
Patron	Walking
	Stopped
	Sitting
Gerontological Centers	Dining Area
	Sleeping Area
	Waiting room Area
	Bathroom Area
	Patio Area
	Garden Area
	Physical Therapy Area
	Manualidades Area
	Recreation Area
	Electrotherapy Area
	Street Area
	Corridor Area
Reception/Cognition-Attention	Orientation
	Sensation/Perception
	Cognition
	Communication
Activity/Rest-Rest/Sleep	Activity/Exercise
	Energy balance
	Cardiovascular/Pulmonary Response
	Self-care
Nutrition	Ingestion
	Digestion
	Absorption
	Metabolism
	Hydration

**Table 2 healthcare-11-01392-t002:** Relationship of classes.

Pattern		Large Class Status
Walking	Relation	Disoriented
Standing		Nervous
Sitting		Bored
		Wanders
		Depressed

**Table 3 healthcare-11-01392-t003:** The table depicts the patients’ moods together with their concepts.

Mood	Concept
Disoriented	Disorientation is conceptualized in the context of the study and based on the researcher’s real observations as the process through which an individual loses their capacity to recognise places, familiar faces, or is confused with their position.
Nervous	In research, anxiousness is conceptualized as the patient’s repetitive acts, such as the formulation of queries, dialogues on the same topic, and tasks already completed, such as eating breakfast. When these actions are not completed, patients experience worry, which leads to anxiousness.
Bored	Boredom is associated with social leisure and demotivation activities, as well as a lack of interest in carrying out activities in recreational spaces, according to the study; based on the above, it is understood that boredom consists of a reluctance to do the things that an individual has traditionally done. The following actions are seen and signify boredom in this context: impulse to read a book, look at images, growth of physical and creative activity
Wandering	For the research process, the state of mind regarding wandering was related to a walk without direction, sense, and spatial orientation of the people observed with Alzheimer’s.
Depressed	Depression is a mental state conceptualized by the investigation as a loss of capacity for the growth of motor activities as well as a deterioration of memory for the execution of voluntary acts.

**Table 4 healthcare-11-01392-t004:** Important elements of the primitives.

Time	Pose	Status
0:00:1	1	Wander
0:00:2	2	Wander
0:00:3	3	Nervous
0:00:4	1	Depressed
0:00:5	1	Depressed
…	…	D…
0:00:55	2	Depressed
0:00:56	2	Depressed
0:00:57	2	Nervous
0:00:58	1	Nervous
0:00:59	1	Depressed
0:00:60	1	Depressed

## Data Availability

In GITHUB there is a repository called Ontologia-Alzheimer where information that justifies the study can be found. Link: https://github.com/Castle73/Ontologia-Alzheimer (accessed on 29 March 2023).

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
