# Peer review of "Ontological Model in the Identification of Emotional Aspects in Alzheimer Patients"

_healthcare, 2023, doi:10.3390/healthcare11101392_

Round 1
Reviewer 1 Report
This paper describes the development of a conceptual representation or ontology related to Alzheimer’s patients. Authors mention that the ontology is intended to clarify semantic context of Alzheimer patient’s relation with their behavior.
The topic is interesting, however there are several aspects that need to be included or improved in the paper to clarify the research relevance and contribution.
1 It is not clear to this reviewer how the hierarchy of classes presented in table 1 and table 3 was developed. It is important to include in the paper if these classes were proposed by physicians or previous works and in that case include the corresponding references.
2 Table 3 and table 4 are not mentioned in the manuscript. Table 3 represents a set of moods, it is not clear how they were defined and why this subset.
3 The authors did not include an Evaluation section, which is required to validate the contributions.
The conclusions are a list of steps or process, not the work’s conclusions. This section needs to be improved and clarify if the results were as expected, what was improved and what is missing. This section should be related to the evaluation section.
5 Finally it is not clear how the proposed ontology would be used to clarify the context and behavior relation as stated in the introduction. It is important to add a section where this is explained.
Author Response
Se publican los cambios sugeridos

Reviewer 2 Report
The present work describes the development of a conceptual representation model of the domain of the theory of formal grammars and abstract machines through ontological modeling. The main goal is to develop an ontology capable of deriving new knowledge about the mood of an Alzheimer’s patient in the categories of wandering, nervous, depressed, disoriented or bored. The patients are from elderly care centers in Ambato Canton - Ecuador. The population consists of 39 individuals of both sexes, diagnosed with Alzheimer’s disease, with ages ranging from 75 to 89 years. The methods used are the taxonomic levels, the semantic categories and the ontological primitives.
I found that the paper is well-written and interesting. Some point need to be fixed.
1- The authors' affiliations are not written correctly.
2- The plagiarism check-up is high 27%
3- The structure of the paper need to be modified as there are a big paragraphs.
4- The aim of this paper needs to be clarified as bullet points
5- The table 1 and 2 header need to be smaller.
overall all paper is good
Author Response
Se publican los cambios sugeridos

Reviewer 3 Report
The authors describe a method to develop the ontology of Alzheimers' disease (AD) symptoms.
The goal of the work is to establish a formal communication between the behaviorof AD patients, as they are observed (in this case) via video recording, and any more-or-less semi-automated systems involved in the study of the disease, as well as (but this is readers' speculation) in care giving.
The chain of different steps going from the observables and tags towards the interaction with machine-learning tools (downhill to ontology) is summarized in Fig.1 and roughly described in Section 2.
The reader can not easily assess what is really new in the manuscript compared to the previous works mentioned by the authors in the manuscript (refs. 21-39, all dealing with AD).
It is clear that the development of the formal ontology is the focus.
But it is also clear that such ontology is present in the previous works too, maybe under a different and likely less structured formal representation.
The aim of the authors is, according to Introduction, not new.
The authors (end of Section 1, lines 145-156) do not stress the novelty of their study.
The work described in the manuscript is limited to develop strict logic (hierarchy, classification, ordering, semantics, etc.), while at present fuzzy logic and natural languages seem to perform better in the AI field.
The possibility of a feedback (typical of deep learning and of methods based on partial learning by experience) is not clearly introduced as a modern methodology.
Indeed, reference 5 (2009) is not recent (line 61).
This issue is particularly important in AD monitoring, because of the elusive signs related to early (but important) mild cognitive impairment.
As for reliability, including non-AD subjects in the study can be a useful test.
The manuscript is basically a tutorial of Protege applied to AD ontology development.
Even though the application described in the manuscript can be of interest to Healthcare's readers, the tutorial is not well written.
Section 2 and lines 406-407 do not provide enough information to exploit AD as a reproducible tutorial.
Searching for "Alzheimer" in Github one can find 25 repositories: indications are not complete.
The application field of the computational chain where ontology is inserted is not completely clear.
Lines 157-163 - The manuscript structure is not that described within these lines.
There are many issues with the text structure.
Line 234 - There is no Fig. 5.
Line 249 - The description of Fig. 6 is not appropriate.
Sentences can not be understood.
Line 262 - There is no Fig. n.
The language of the manuscript requires a careful check:
sentences are sometimes repeated, truncated, can not be understood, etc..
Author Response
Se publican los cambios sugeridos

Round 2
Reviewer 1 Report
Authors have improved the paper according to review 1. The article is difficult to read in some paragraphs, including the introduction and paragraph 228-232, which is not clear, it needs to be presented in a separate section an extend the explanation.
Author Response
The file with the corrected observations is sent.

Reviewer 2 Report
This is a review for this paper that is targeting to:
The methods used are the taxonomic levels, the semantic categories and the ontological primitives. 7 All these aspects allow the computational generation of an ontological structure, in addition to the use 8 of the proprietary tool Pellet Reasoner as well as Apache NetBeans from Java for process completion. 9 As a result, an ontological model is generated using its instances and Pellet Reasoner to identify the 10 expected effect. It is noted that the ontologies come from the artificial intelligence domain. In this 11 case, they are represented by aspects of real-world context that relate to common vocabularies for 12 humans and applications working in a domain or area of interest.
1- the affiliations of the authors are not provided.
2- Introduction and literature review have a lot of similarities and need to be paraphrased.
Author Response

(The authors gave the same response as above.)

Reviewer 3 Report
The authors only partially replied to the points raised by this reviewer.
The language and the presentation of the work have been improved significantly. The authors should split long sentences. For instance in Conclusions there is a sentence of 8 lines.
Questions 5-7 have been adequately answered.
Questions 8-9 (detect early signs and including non-AD as for comparison) have not been understood by authors. The questions deal with the possibility to artificially detect early symptoms of AD and to distinguish them from the background of non AD observations. I do not see in the manuscript any demonstration of such capability by AI, so far.
The main issue is still about the novelty of the approach. Lines 169-172 do not answer questions 1-4, because it is not clear what is new compared to all of the studies listed in Introduction. The authors should make a little effort to stress what is new, before claiming to be "groundbreaking".
Finally the "Data Availability Statement" disappeared, thus affecting the reproducibility (question 11).
Author Response

(The authors gave the same response as above.)
